# In Vivo PET Detection of Lung Micrometastasis in Mice by Targeting Endothelial VCAM-1 Using a Dual-Contrast PET/MRI Probe

**DOI:** 10.3390/ijms25137160

**Published:** 2024-06-28

**Authors:** Stavros Melemenidis, James C. Knight, Veerle Kersemans, Francisco Perez-Balderas, Niloufar Zarghami, Manuel Sarmiento Soto, Bart Cornelissen, Ruth J. Muschel, Nicola R. Sibson

**Affiliations:** 1Department of Radiation Oncology, Stanford School of Medicine, Cancer Institute, Stanford University, Stanford, CA 94305, USA; stavmel@stanford.edu; 2School of Natural and Environmental Sciences, Newcastle University, Newcastle upon Tyne NE1 7RU, UK; james.knight2@newcastle.ac.uk; 3Clinical Nuclear Medicine Imaging, Siemens Healthineers, 2595 BN The Hague, The Netherlands; veerle.kersemans@siemens-healthineers.com; 4CureVac, AG. Paul-Ehrlich-Str. 15, 72076 Tubingen, Germany; franciscoperezbalderas@gmail.com; 5Department of Oncology, University of Oxford, Oxford OX3 7DQ, UK; nili.zarghami@gmail.com (N.Z.); ruth.muschel@oncology.ox.ac.uk (R.J.M.); 6Department of Biochemistry and Molecular Biology, University of Seville, 41004 Seville, Spain; manuelsarmientosoto@gmail.com; 7Department of Nuclear Medicine, University Medical Center Groningen, Hanzeplein 1, 9713 GZ Groningen, The Netherlands; bart.cornelissen@oncology.ox.ac.uk

**Keywords:** micrometastasis, early detection, microparticles of iron oxide (MPIO), vascular cell adhesion molecule-1 (VCAM-1), dual modality (PET/MRI)

## Abstract

Current clinical diagnostic imaging methods for lung metastases are sensitive only to large tumours (1–2 mm cross-sectional diameter), and early detection can dramatically improve treatment. We have previously demonstrated that an antibody-targeted MRI contrast agent based on microparticles of iron oxide (MPIO; 1 μm diameter) enables the imaging of endothelial vascular cell adhesion molecule-1 (VCAM-1). Using a mouse model of lung metastasis, upregulation of endothelial VCAM-1 expression was demonstrated in micrometastasis-associated vessels but not in normal lung tissue, and binding of VCAM-MPIO to these vessels was evident histologically. Owing to the lack of proton MRI signals in the lungs, we modified the VCAM-MPIO to include zirconium-89 (^89^Zr, *t*_1/2_ = 78.4 h) in order to allow the in vivo detection of lung metastases by positron emission tomography (PET). Using this new agent (^89^Zr-DFO-VCAM-MPIO), it was possible to detect the presence of micrometastases within the lung in vivo from ca. 140 μm in diameter. Histological analysis combined with autoradiography confirmed the specific binding of the agent to the VCAM-1 expressing vasculature at the sites of pulmonary micrometastases. By retaining the original VCAM-MPIO as the basis for this new molecular contrast agent, we have created a dual-modality (PET/MRI) agent for the concurrent detection of lung and brain micrometastases.

## 1. Introduction

Metastasis from a primary tumour site to distant organs is a complex succession of events collectively known as the invasion-metastasis cascade [1]. Currently metastasis accounts for 90% of human cancer deaths and is one of the major challenges in cancer therapy [2]. Owing to the capacity of cancer cells to spread to the lungs, this is the second most frequent site of metastasis, with an estimated 20 to 54% of tumours originating elsewhere in the body metastasising to this organ [3]. The most prevalent cancers that metastasise to the lung parenchyma include those originating from the breast, lung, colon, uterine leiomyosarcoma, and head and neck squamous cell carcinomas [3,4]. In children, it is rare to see primary lung cancer, and the majority of lung cancer metastasis cases metastasise from a primary site to the lungs [5]. The most frequently employed non-invasive imaging modalities for identifying lung metastases are computed tomography (CT) with or without the application of contrast agents and ^18^F fluorodeoxyglucose (^18^F-FDG) positron emission tomography (PET)/CT scans. Although the improvement in CT technologies allows for spatial resolution detection of lung nodules below 5 mm, it cannot provide criteria to predict the nature of these nodules [6]. ^18^F-FDG PET complements the CT resolution, offering information about uncertain small lesions and their metabolic activity. It is important to note, however, that the detection sensitivity of ^18^F-FDG PET relies on the accumulation of highly metabolically active cells (both proliferating tumour cells and inflammatory cells) to allow tumour detection/diagnosis, which typically reflects a relatively late stage of tumour development and leaves micrometastases undetected [7,8]. This resolution limitation results in decreased sensitivity for low-volume diseases, leading to potential false negatives [9,10]. Moreover, CT scans depend on size and density contrast to identify abnormalities and such contrasts may not be evident in early-stage disease, causing micrometastases to be overlooked [11]. At the same time, ^18^F-FDG is not exclusive to tumours and accumulates in any glucose-avid tissue, including those affected by inflammation. This non-specific uptake can obscure the presence of a disease or mimic it, complicating the differentiation between cancerous and non-cancerous conditions [12,13]. Owing to the lack of imaging sensitivity for the early stages of tumour cell extravasation from circulation and micrometastatic formation, clinical therapy is often applied at a time that is beyond the point of effective intervention. Thus, significant advantages may be gained if more sensitive imaging methods can be developed to allow molecular targeting of markers that enable the early detection of micrometastatic colonies.

We have previously demonstrated that it is possible to detect micrometastases in mouse brain very early in their development using molecularly targeted magnetic resonance imaging (MRI) via the associated endothelial activation and upregulation of endovascular adhesion molecules [14,15,16]. These endothelial adhesion molecules are rapidly upregulated in response to disease or injury and mediate leukocyte rolling, adhesion, and transmigration across the vascular wall [17]. There is now evidence to suggest that tumour cells also use such adhesion molecules to promote their adhesion to and extravasation across the vascular endothelium [18]. One such adhesion molecule is vascular cell adhesion molecule 1 (VCAM-1), which is upregulated very early in brain metastasis formation [19] and has also been shown to enhance endothelial cell recruitment for new blood vessel formation, thus supporting tumour growth and spread [20]. The early and marked upregulation of VCAM-1 makes this molecule a potential early biomarker for detecting metastatic activity, offering advantages over traditional imaging markers that may not clearly distinguish between tumour stages or identify early metastatic spread. To this end, using an MRI contrast agent based on microparticles of iron oxide (MPIO) and targeted to VCAM-1, we have previously demonstrated that VCAM-1 provides a sensitive and specific endothelial biomarker of early brain metastasis [14,15], as well as improving detection of the tumour-brain interface [21].

Previous studies have also shown that VCAM-1 is markedly upregulated in both lung and liver metastases [22,23], with little expression in the normal lung endothelium. Thus, VCAM-1 potentially represents a target for the early detection of micrometastasis in the lungs. However, owing to the low proton density of the lung tissue, it is not possible to effectively detect the presence of MPIO by MRI in the lung, as this relies on dephasing existing proton signals to achieve binding-specific contrast. Alternatively, positron emission tomography (PET), which is typically combined with CT, offers an attractive solution as it provides high sensitivity (10^−12^ M), enabling the location and distribution of radiolabelled molecules in the lung to be tracked in vivo. The prevailing PET imaging agent for identifying pulmonary metastases is ^18^F-FDG, which accumulates in regions with elevated glucose metabolism. This functional imaging method, however, is size-limited to tumours greater than 1 cm in diameter [24,25]. In the current study, therefore, we propose that labelling the VCAM-MPIO contrast agent with a positron-emitting radioisotope such as ^89^Zr will enable PET detection of micrometastases in the lung. In prior studies, PET probes featuring different radioisotopes directed towards VCAM-1 have been reported, utilising either nanobodies derived from unique heavy-chain-only antibodies [26] or antibodies alone [27]. However, these probes have very long circulation times in the blood and are also likely to extravasate from the vasculature. Our approach maintains all the advantages of the MPIO-based strategy, including short circulatory half-life and high valency for targeting ligands, while enabling detection in an organ that would be insensitive to the molecular MRI approach. Further, this novel agent would enable sequential or concurrent MRI and PET detection of micrometastases in different organs (e.g., the brain and lungs) within the same subject.

The aim of this study, therefore, was to develop a dual-contrast PET/MRI probe targeting VCAM-1 to enable the early detection of micrometastases in both the brain and lungs using a single contrast agent. This dual-modality approach not only promises to improve the accuracy of metastasis detection but may also facilitate the early initiation of targeted therapies, potentially improving patient outcomes significantly.

## 2. Results

### 2.1. Non-Radiolabelled VCAM-, IgG- or BSA-MPIO Conjugates

#### 2.1.1. Synthesis and Loading of Conjugates

Both VCAM-MPIO and IgG-MPIO were successfully synthesised, and the antibody loading was calculated to be ca. 3350 antibodies per μm^2^ of the MPIO surface (11,650 per MPIO). Following the ^89^Zr radiolabelling reactions, approximately 250 ^89^Zr atoms were associated with each MPIO (MA = 1.06 × 10^−4^ MBq/μg of MPIO).

#### 2.1.2. Ex Vivo Quantitation of VCAM-MPIO Binding in Pulmonary Metastasis Model

Adherent VCAM-MPIO were evident in metastasis-associated microvasculature and nearby larger vessels in the lungs of mice injected with VCAM-MPIO (Figure 1A,B; Appendix A), while few or no MPIO were found in the lungs of naïve mice injected with VCAM-MPIO. Specific binding of VCAM-MPIO on marked VCAM-1 expressing lung vasculature was confirmed immunohistochemically (Figure 1A,B) and, again, was absent from the lungs of naïve mice. Studies in an initial cohort of animals (n = 6) demonstrated the reproducibility of the lung micrometastasis model used, validating subsequent comparisons between groups; no significant differences were found in either tumour number (4.50 ± 1.23; mean number of metastases per lung section analysed ± standard deviation [SD]) or area-derived diameter (137.01 ± 45.17 μm; mean ± SD) between animals (Appendix A).

A significantly greater number of VCAM-MPIO than IgG-MPIO (quantified as the number of adherent MPIO per μm of inflamed endothelium, excluding capillaries) were found associated with micrometastatic colonies at 10 min post-injection [VCAM-MPIO vs. IgG-MPIO, (9.80 ± 1.35) × 10^−3^ vs. (4.45 ± 4.08) × 10^−4^ MPIO/μm endothelium; mean ± SD; unpaired *t* test *p* < 0.005; Figure 1C]. VCAM-MPIO retention in metastasis-bearing mice was significantly reduced at 3 h compared to metastasis-bearing mice at 10 min post-injection [(1.54 ± 1.82) × 10^−3^ vs. (9.8 ± 1.35) × 10^−3^ MPIO/μm^2^ tumour area; mean ± SD; *p* < 0.01; Figure 1C], most likely owing to immune cell (neutrophil) phagocytosis (Appendix A).

Quantitative analysis of non-specific MPIO retention assessed from normal tissue (i.e., not containing metastases) in the lungs of 4T1-GFP injected animals showed substantially lower levels than for metastasis-associated retention at 10 min post-injection ((2.51 ± 1.07) × 10^−5^ MPIO/μm^2^ vs. (0.12 ± 0.09) × 10^−2^ MPIO/μm^2^, *p* < 0.0001; Figure 1D). Similarly, low levels of retention were found in the naïve mice injected with either VCAM-MPIO ((1.22 ± 0.05) × 10^−5^ MPIO/μm^2^) or IgG-MPIO ((0.64 ± 0.21) × 10^−5^ MPIO/μm^2^).

### 2.2. Radiolabelled VCAM-, IgG- or BSA-MPIO Conjugates

#### In Vitro Binding of [^89^Zr]-DFO-VCAM-MPIO

Radioimmunoassays were performed to confirm that the binding characteristics of VCAM-MPIO were not affected by radiolabelling, and indicated significantly greater binding of [^89^Zr]-DFO-VCAM-MPIO to recombinant mouse VCAM-1 (% binding ± SD, 29.51 ± 7.03; Figure 2A) than both of the negative controls [^89^Zr]-DFO-IgG-MPIO (10.34 ± 0.57; *p* < 0.001) and [^89^Zr]-DFO (13.41 ± 1.20; *p* < 0.005). No significant difference was found between the binding of [^89^Zr]-DFO-VCAM-MPIO and the positive control [^89^Zr]-DFO-VCAM (22.78 ± 2.51). A significant reduction in binding was observed following partial blocking with VCAM-1 antibody (% cpm ± SD, 14.30 ± 2.28; *p* < 0.001 compared to unblocked binding; Figure 2A).

Subsequently, binding to endothelial VCAM-1 was assessed on TNF-stimulated HUVECs and, again, [^89^Zr]-DFO-VCAM-MPIO showed significantly greater binding (% ± SD, 14.23 ± 4.43; *p* < 0.01; Figure 2B) than either of the negative controls [^89^Zr]-DFO-IgG-MPIO (4.22 ± 0.92) and [^89^Zr]-DFO (3.27 ± 0.72). In contrast, the positive control [^89^Zr]-DFO-VCAM showed similar levels of binding (17.69 ± 4.71) to [^89^Zr]-DFO-VCAM-MPIO. Pre-incubation with an excess of unlabelled VCAM-1 antibody effectively out-competed binding of both [^89^Zr]-DFO-VCAM-MPIO and [^89^Zr]-DFO-VCAM to TNF-stimulated HUVECs (% cpm ± SD, 4.45 ± 0.94 and 6.93 ± 0.99, respectively; Figure 2B). The binding of [^89^Zr]-DFO-VCAM-MPIO to naïve unstimulated HUVECs was negligible (1.59 ± 0.41; Figure 2B).

### 2.3. In Vivo Imaging

#### 2.3.1. In Vivo PET Imaging of [^89^Zr]-DFO-VCAM-MPIO in Pulmonary Metastasis Model

The binding of [^89^Zr]-DFO-VCAM-MPIO in vivo in mice injected with 4T1-GFP cells was compared to that of both [^89^Zr]-DFO-IgG-MPIO and the agent without MPIO, [^89^Zr]-DFO-VCAM, at 10 min post-administration using PET. In accord with the known route of MPIO clearance, the highest levels of activity for both [^89^Zr]-DFO-VCAM-MPIO (Figure 3A and Appendix A) and [^89^Zr]-DFO-IgG-MPIO (Figure 3B and Appendix A) were found in the liver and spleen. Although radioactivity within the lungs was not visually apparent on individual single-slice static PET images from pulmonary metastasis-bearing mice injected with [^89^Zr]-DFO-VCAM-MPIO (Appendix A), maximum intensity plots obtained from the full 3D dataset revealed marked radioactive uptake in the lungs (Figure 3A) compared to metastasis-bearing mice injected with the isotype control agent [^89^Zr]-DFO-IgG-MPIO (Figure 3B). Although radioactivity was clearly evident in the 3D maximum intensity plots from the diseased lungs, this could not be resolved into individual micrometastatic colonies (Figure 3A), likely owing to a combination of movement and the relatively low spatial resolution of the PET images.

Naïve (control) mice injected with [^89^Zr]-DFO-VCAM-MPIO also showed uptake, primarily in the liver and spleen (Appendix A). Metastasis-bearing animals injected with [^89^Zr]-DFO-VCAM (antibody alone) showed high levels of the agent in the blood for the first hour (Appendix A) and high levels of radioactivity in the heart. The liver and bladder also showed high uptake of radioactivity, whereas lower levels of radioactivity were evident in the spleen and bones (Appendix A).

Despite the lack of visible focal signal in the lungs of metastasis-bearing animals injected with [^89^Zr]-DFO-VCAM-MPIO, time-activity curves for the lungs showed clear activity uptake (equilibrating to 0.53 ± 0.28 g/mL; mean ± SD; Figure 3C,E). This uptake was significantly higher than the activity uptake of either [^89^Zr]-DFO-IgG-MPIO in lungs from metastasis-bearing mice (0.12 ± 0.04 g/mL; mean ± SD; *p* < 0.0001; Figure 3D,F) or [^89^Zr]-DFO-VCAM-MPIO uptake in naïve mouse lungs (0.10 ± 0.02 g/mL; mean ± SD; *p*< 0.0001; Figure 3G). The blood activity in all groups was consistently lower than that of lung uptake of [^89^Zr]-DFO-VCAM-MPIO in metastasis-bearing animals (Figure 3C–G; *p* < 0.0001).

Time-activity curves for the livers of metastasis-bearing animals injected with [^89^Zr]-DFO-VCAM-MPIO (Figure 3C) or [^89^Zr]-DFO-IgG-MPIO (Figure 3D) showed a rapid increase in uptake, which plateaued at 5 min post-administration. Metastasis-bearing mice injected with [^89^Zr]-DFO-VCAM-MPIO showed comparable levels of liver uptake, over 50 min, to those injected with either [^89^Zr]-DFO-IgG-MPIO (2.65 ± 0.28 vs. 2.38 ± 0.12 g/mL; mean ± SD; Figure 3E,F) or naïve mice injected with [^89^Zr]-DFO-VCAM-MPIO (2.52 ± 0.32 g/mL; mean ± SD; Figure 3E,G).

Activity uptake curves for [^89^Zr]-DFO-VCAM in metastasis-bearing mice showed a rapid increase in uptake into the heart, liver, and bladder (Appendix A), reaching a plateau at 5 min post-administration. The half-life of [^89^Zr]-DFO-VCAM in the circulation was determined from the heart measurements. Heart measurements were also used to determine the half-life of [^89^Zr]-DFO-VCAM-MPIO or [^89^Zr]-DFO-IgG-MPIO in the circulation, and findings were consistent with the values calculated from the biological clearance study; circulatory half-life PET vs. MRI, 36.6 vs. 39.9 s (Appendix A).

Static analysis of lung activity was also performed at two time points, 10 and 55 min post-administration, on the 1 h in vivo dynamic dataset. Lungs from the metastasis-bearing group injected with [^89^Zr]-DFO-VCAM-MPIO showed significantly greater (*p* < 0.0005) activity compared to both metastasis-bearing animals injected with [^89^Zr]-DFO-IgG-MPIO and naïve animals injected with [^89^Zr]-DFO-VCAM-MPIO at both time points (Figure 4A,B).

#### 2.3.2. Post Mortem PET Imaging of [^89^Zr]-DFO-VCAM-MPIO in Pulmonary Metastasis Model

To circumvent issues of motion in the in vivo PET images, two metastasis-bearing mice injected with either [^89^Zr]-DFO-VCAM-MPIO (Figure 4C) or [^89^Zr]-DFO-IgG-MPIO (Figure 4D) were terminated 10 min post-administration and imaged over a 2 h time frame with PET. Both images showed high levels of radioactivity in the liver, while only the mouse injected with [^89^Zr]-DFO-VCAM-MPIO showed significant radioactivity in the lungs. Subsequently, the mouse injected with [^89^Zr]-DFO-VCAM-MPIO was further investigated with 60 h PET acquisition after excision of all organs below the diaphragm to verify that the signal originated from the lungs and was not a partial volume effect from the liver and spleen (Figure 4E).

### 2.4. Ex Vivo Analysis

#### 2.4.1. Ex Vivo Biodistribution of [^89^Zr]-DFO-VCAM-MPIO

Biodistribution of all agents was further assessed ex vivo, and the highest uptake across all of the organs studied was found in the lungs from metastasis-bearing mice injected with [^89^Zr]-DFO-VCAM-MPIO (68.9 ± 22.9% ID/g; mean ± SD); lung uptake was significantly higher than that in all other experimental groups (one-way ANOVA, *p* < 0.05; Figure 5). The spleen and liver exhibited the second highest activity uptake in metastasis-bearing mice injected with [^89^Zr]-DFO-VCAM-MPIO (49.6 ± 37.6% ID/g; mean ± SD), and the levels were similar in all groups. The heart, kidney, bone, and blood samples showed relatively low levels of uptake, while the stomach, small and large intestine, pancreas, and samples from muscle, skin, and fat showed negligible uptake in all groups (Figure 5).

#### 2.4.2. Lung Autoradiography of [^89^Zr]-DFO-VCAM-MPIO Accumulation in Lungs

The binding of [^89^Zr]-labelled MPIO was further confirmed by comparison of autoradiography with immunohistochemical assessment of VCAM-1 expression and the presence of metastases in the same sections (Figure 6). In all sections from metastasis-bearing mouse injected with [^89^Zr]-DFO-VCAM-MPIO, a low level of activity was evident across the whole lung, in addition to more intense dark foci in discrete areas of the lungs (Figure 6A). The negligible activity was evident in sections from either metastasis-bearing or naïve mice injected with [^89^Zr]-DFO-IgG-MPIO (Figure 6C,D) and from the naïve mouse injected with [^89^Zr]-DFO-VCAM-MPIO (Figure 6B). Lung tissue sections were also used for longer exposure autoradiography and subsequent VCAM-1 immunohistochemistry, and the presence of metastases at the location of the intense dark spots on the autoradiography film was confirmed; these further correlated with the presence of [^89^Zr]-DFO-VCAM-MPIO (Figure 6E–G; Appendix A). At longer exposures, it was apparent that some dark spots on the film were not correlated with metastases but were correlated with the presence of MPIO (Appendix A; small red dots on the tissue). These MPIO may be associated with metastases in subsequent tissue sections or may reflect non-specific retention owing to vessel/tissue architecture. It was also found that occasional metastases with a very low MPIO score did not correlate with foci (Appendix A), most likely owing to low ^89^Zr levels. Conversely, in a small number of metastases showing high expression of VCAM-1, it appeared that no MPIO were present, but these still correlated with moderate activity foci (Appendix A). In this case, it is possible that the MPIO were dislodged during the staining process or obscured by the very intense VCAM-1 staining. The autoradiography in Figure 6E was performed a day earlier than that in Figure 6A and was exposed for 24 h compared to 10 h for the sections in Figure 6A. Because the exposure time was not proportional to the decay of the isotope, the autoradiograph in Figure 6E appears darker than the samples in Figure 6A, due to overexposure. Prior to autoradiography, the lungs were not perfused with saline to remove the blood from the vasculature, and it is likely, therefore, that the uncorrelated dark spots reflect an unknown amount of free ^89^Zr in the blood. This notion is supported by the higher signal observed in the heart, which contains a greater blood volume than lung tissue. It should also be noted that in some areas where the tissue is folded, a higher saturated signal is evident that does not reflect the presence of metastases (Figure 6E,F; Appendix A).

## 3. Discussion

To date, there is no standard non-invasive clinical or preclinical screening method that provides a diagnosis for early micrometastatic development in the lung. In this study, we show that [^89^Zr]-DFO-VCAM-MPIO specifically binds to activated pulmonary endothelium at sites of micrometastases, enabling very early detection of lung metastases by PET imaging.

The spread of metastatic cells to the lymphatic system is common in many types of cancer. Consequently, imaging of lymph nodes has become one of the standard clinical screening methods for secondary tumour spread, and both structural MRI [28] and nuclear imaging with functional tracers [29,30] have enabled the detection of malignant lymph nodes for many years.

Contrast-enhanced MRI using superparamagnetic particles of iron oxide (SPIO; 40–150 nm) and ultrasmall SPIO (USPIO; <50 nm) have been utilised in the clinic for their properties to preferentially phagocytosed by cells in normal lymphoid tissues [31] and facilitated a more accurate assessment of lymph node malignancy [32,33,34,35,36]. However, these methods are limited to the detection of macro- and micrometastatic lesions within the lymphatic system and do not enable site-specific identification of spread to other tissues/organs [16].

Targeting of VCAM-1 with VCAM-MPIO and in vivo MRI has previously been demonstrated in preclinical models of numerous diseases, including neuroinflammation [37], cardiovascular disease [38], multiple sclerosis [39], epilepsy [40], and, most recently, brain metastasis [14,15,21]. Based on the same VCAM-MPIO agent, a recent study has demonstrated the use of radiolabelled VCAM-MPIO for bimodal (SPECT and MRI) imaging of brain inflammation in rats [41]. VCAM-1 has also been shown to be upregulated in pulmonary endothelial cells early in the development of metastatic foci in a biphasic manner [22]. In the current study, we demonstrated, using ex vivo immunohistochemistry, that VCAM-MPIO specifically binds to activated pulmonary vessels expressing VCAM-1, which are closely associated with metastases, with minimal non-specific binding in either metastasis-bearing or naïve mice. We have demonstrated that radiolabelling of VCAM-MPIO does not affect binding specificity, as previously shown [14,15], and that binding of [^89^Zr]-DFO-VCAM-MPIO at sites of metastasis in the lungs can be detected in vivo by PET imaging. This approach has enabled the presence of metastases at the micrometastatic stage, with a mean diameter of 140 μm, to be detected.

This novel PET/MRI contrast agent showed rapid clearance from the blood circulation (<40 s), primarily to the liver and spleen, in accordance with previous work [42]. Micrometastasis-bearing lungs could be identified both in maximum intensity plots from the full 3D PET datasets and also by volumetric quantitation of radioactivity within the first 10 min after administration of the [^89^Zr]-DFO-VCAM-MPIO contrast agent. Comparable radioactivity levels in the lungs were observed throughout the 1 h dynamic PET acquisition. It was not possible to differentiate the signal within the lungs into individual sites of micrometastasis, likely reflecting the effects of movement, since PET imaging was performed without respiratory or cardiac gating, and the relatively low resolution of PET imaging compared to the size of the micrometastatic colonies. For this reason, we analysed lung radioactivity uptake as the cumulative volumetric effect from many metastases across the whole lung and were able to visualise this global uptake in the maximum intensity PET images from the lungs. Sequential autoradiography and immunohistochemical staining of lung tissue confirmed co-localisation of radioactivity signals from [^89^Zr]-DFO-VCAM-MPIO and VCAM-1 staining on micrometastasis-associated vasculature, indicating the specific binding of our probe.

To confirm the specificity of lung uptake of [^89^Zr]-DFO-VCAM-MPIO in metastasis-bearing lungs, we acquired 2 h post-mortem PET images, enabling acquisition free of motion artefacts. These images showed a marked signal in the lungs of metastasis-bearing mice injected with [^89^Zr]-DFO-VCAM-MPIO, which was not evident in metastasis-bearing mice injected with [^89^Zr]-DFO-IgG-MPIO. Subsequently, a longer post-mortem PET acquisition (60 h) of a metastasis-bearing mouse injected with [^89^Zr]-DFO-VCAM-MPIO was performed following the removal of all organs from the diaphragm. These images confirmed that the source of radioactivity was the lungs and eliminated the possibility of partial volume effects from the liver and spleen.

The rapid biological clearance of VCAM-MPIO from the blood ensures a low background signal and enables the detection of target-specific binding. The measured rates of clearance (*t*_1/2_ = 39.9 s) were comparable to those of other, slightly larger, 2.8 μm diameter MPIO (*t*_1/2_ = 34.9 s) reported previously [42] and much shorter than that measured for ultrasmall iron oxide nanoparticles (20 nm; *t*_1/2_ = 2.6 h), which have been used in a previous molecularly targeted MRI study [42]. These differences highlight one of the key ways in which nanoparticle-sized platforms are inappropriate for short-lived isotope imaging probes owing to the persistent blood pool background. In contrast, micron-sized platforms provide a class of particles that have a large surface for polyvalent ligand loading and favourable blood clearance characteristics for nuclear medicine imaging probes with short-lived isotopes. Further, by retaining the MRI-detectable MPIO platform, we provide a dual-modality contrast agent, which could be used to detect both brain metastases (by MRI) and lung metastases (by PET) within the same individual, utilising the higher spatial resolution afforded by MRI in the brain, where the avoidance of treatment damage to normal tissue is of even greater importance. However, since it was necessary to use a lower MPIO dose in the current study (1.6 mg/kg Fe) than that previously used to detect brain metastases by MRI (4 mg/kg Fe) [14], we assessed the sensitivity of this lower dose in a mouse model of brain metastasis. Detection of micrometastases was found to be equally sensitive as with the higher dose used previously (Appendix A) and in the context of additional binding at sites of lung metastases (Appendix A). Thus, the new bimodal probe can be used at concentrations appropriate for lung micrometastasis detection by PET imaging and still enables brain micrometastasis detection by MRI. The ability to obtain such information about metastatic spread could transform the management of a significant number of patients with cancer. Moreover, this concept provides a new, and possibly the first, compelling argument for dual-modality PET/MRI imaging probes for organ-specific micrometastases detection.

It is important to note that at the 3 h mark, as illustrated in Figure 1C, a notable decrease in the specific binding retention was evident, which we believe can be attributed to the phagocytic action of innate immune cells clearing bound MPIO. In preliminary studies using a higher concentration of VCAM-MPIO, phagocytosis of MPIO by intravascular neutrophils was observed as early as 1 h after injection (Appendix A). This phagocytic activity was not observed at 1 h with the lower dose of VCAM-MPIO used in the remainder of this study (Appendix A) but is a likely explanation for the subsequent clearance of bound MPIO. Considering that PET scans can be initiated as swiftly as 10 min after the radiotracer is administered, potentially within the scanner itself, concerns regarding phagocytosis primarily emerge in relation to the subsequent MRI of VCAM-MPIO binding in the brain. However, unlike the lung microvasculature, cerebral vessels do not contain a high number of neutrophils; thus, the likelihood of intraluminal phagocytosis confounding subsequent measurement of VCAM-MPIO binding in the brain is greatly reduced. Moreover, previous studies using the VCAM-MPIO construct in brain inflammation models demonstrated consistent MRI contrast for up to 5 h after the onset of binding [34].

Another issue that must be considered for the clinical application of this approach is that CAMs, including VCAM-1, may be upregulated in the vasculature by inflammatory processes unrelated to metastasis. Nevertheless, in the context of “at risk” primary cancer patients, where the assessment strategy would be specifically targeted, the probability that VCAM-1 expression in the lungs reflects metastatic involvement is substantially increased. Moreover, the spatial presentation of VCAM-1 associated with micrometastases, typically manifesting as relatively small hotspots, will differ distinctly from the broader areas of expression typically associated with generalised inflammation. At the resolution of current clinical PET scanners, we would expect these differing spatial patterns of expression, and hence contrast agent binding, to be detectable. Moreover, as PET technology advances, even higher spatial resolutions may become possible, as has been demonstrated preclinically [43]. The primary cancer cohorts with the greatest risk of secondary spread to the lungs are breast, colorectal, kidney, testicular, bladder, melanoma, bone, and soft tissue sarcomas. Thus, periodic screening of patients in these cohorts could be envisaged clinically.

In the context of imaging for early detection of lung micrometastases, our study has demonstrated the potential of a [^89^Zr]-labelled VCAM-MPIO probe. This potential is anchored not in the traditional metric of spatial resolution but in the biological specificity of the tracer to identify early pathological changes associated with micrometastatic disease. In contrast to ^18^F-FDG PET, which relies on downstream functional changes within the tumour microenvironment for detection, the approach described here is based on the very early expression of tumour-specific molecular targets on the tumour-associated vasculature. Consequently, earlier detection could be facilitated. In the current study, the selection of ^89^Zr, with its extended half-life, enabled detailed autoradiographic validation and post-mortem PET imaging.

In considering the future trajectory of imaging technology, the development of an [^18^F]-labelled VCAM-MPIO variant would leverage the superior imaging qualities of ^18^F compared to ^89^Zr, including enhanced spatial resolution as a result of its lower positron range and a reduced radioactivity exposure for patients, owing to its shorter half-life. At the same time, implementing respiratory and cardiac gating would improve sensitivity, potentially enabling individual radiotracer hotspots to be correlated with micrometastases on PET scans. Downstream, co-targeting of additional biomarkers alongside VCAM-1 has the potential to provide a more nuanced view of tumour biology and the microenvironment. Finally, integrating machine learning algorithms with dual-modality imaging data presents an opportunity to refine diagnostic accuracy. Training these algorithms on extensive datasets that yield PET and MRI outputs could foster predictive models that more accurately forecast disease progression and therapeutic responses, leading to a new level of precision in diagnostic imaging.

## 4. Materials and Methods

### 4.1. Pulmonary Metastasis Mouse Model

All animal experiments were approved by the University of Oxford Clinical Medicine Ethics Review Committee and the UK Home Office (Animals [Scientific Procedures] Act 1986) and conducted in accordance with the University of Oxford Policy on the Use of Animals in Scientific Research, the ARRIVE Guidelines, and Guidelines for the Welfare and Use of Animals in Cancer Research [44].

A total number of 96 female 7–10 week-old *BALB/c* mice (Charles River, Oxford, UK) were used for this study. The mice were housed in ventilated cages with a 12 h light/dark cycle and controlled temperature (20–22 °C), fed normal chow, and given water ad libitum. A total of 46 mice were injected intravenously via the tail vein with 5 × 10^4^ 4T1-GFP mouse metastatic mammary carcinoma cells (ATCC, Teddington, UK) in 100 μL sterile PBS and studied at day 10 after tumour cell injection.

### 4.2. Non-Radiolabelled VCAM-, IgG- or BSA-MPIO Conjugates

#### 4.2.1. Synthesis of Conjugates

MyOne™ Tosylactivated MPIO (Thermo Fisher Scientific, Waltham, MA, USA) with 1.08 μm diameter were conjugated to either rat anti-mouse VCAM-1 antibody (M/K 2 clone, Cambridge Bioscience, Cambridge, UK) or rat IgG-1 isotype control antibody (Cambridge Bioscience), as described previously [14]. To obtain BSA-MPIO, the antibody-loading step was performed with a blocking buffer (PBS + 0.5% BSA + 0.05% Tween-20).

#### 4.2.2. Assessment of Antibody Loading

Antibody loading of VCAM-1 or IgG-1 onto MPIO was determined by flow cytometry analysis. Anti-VCAM-1 antibody was fluorescently labelled in a one-step protocol with alexafluor 647 goat anti-rat IgG antibody (cat. Nr. A21247, Thermo Fisher Scientific, Waltham, MA, USA). Qifikit^®^* calibration beads (K0078, Dako, UK) were used as a reference and labelled with alexafluor 647 goat anti-mouse IgG antibody (cat. Nr. A21235, Thermo Fisher Scientific), according to the manufacturer’s protocol. Briefly, 5 μL of VCAM- or IgG-MPIO were diluted in 200 μL of alexafluor 647 goat anti-rat IgG (diluted 1:100 in PBS) and incubated at 23 °C for 30 min. The conjugates were subsequently washed twice with 1 mL PBS containing 0.05% Tween-20 and pelleted on a Dynal magnetic separator (Thermo Fisher Scientific) for 2 min. Finally, the conjugates were redispersed in 500 μL PBS and stored at 4 °C for no more than 1 h prior to flow cytometry. Flow cytometry experiments were performed on a BD FACScalibur (BD Biosciences, Franklin Lakes, NJ, USA) flow cytometer. A calibration curve was constructed according to the manufacturer’s protocol by linear regression of the log-log plot of antibody loading versus fluorescence intensity.

#### 4.2.3. Ex Vivo Assessment of VCAM-MPIO Binding in Pulmonary Metastasis Model

Ex vivo preliminary studies conducted with the dose previously used for the detection of brain metastases (4 mg Fe/kg body weight) showed high non-specific retention of MPIO in the lung microvasculature, likely as a consequence of clogging of particles in the microvasculature owing to its tortuosity. MPIO rapidly sediment out of solution (within 3–5 min) when left to stand. Such sedimentation does not reflect magnetic or chemical forces and is easily separated by vortexing. However, it is likely that if injected at a high concentration, MPIO may clump in highly tortuous microvascular beds, such as those in the lung [45]. However, if a lower-density bolus is injected, the tendency for clogging is markedly reduced and minimised as the bolus density decreases to 40% (1.6 mg of Fe/kg body weight). At this dose, minimal non-specific retention of VCAM-MPIO was observed in the naïve mice. All subsequent experiments reported in this study were performed using an MPIO dose of 1.6 mg Fe per kg body weight. These initial studies also indicated that rapid progression of the 4T1 cell line significantly exacerbates systemic inflammation. Consequently, in mice bearing metastases of >0.5 mm diameter, a marked increase in the number of VCAM-1-positive vessels that were not closely associated with the metastases and a concomitant decline in MPIO retention per tumour area was evident. For this reason, a single early time point, 10 days post-metastasis induction, was chosen to minimise non-tumour related inflammation and enable specific detection of VCAM-1 expression associated with early metastasis development.

Metastasis-bearing mice or naïve animals were injected with 1.6 mg/kg of either VCAM-MPIO or IgG-MPIO via the tail vein in 100 μL saline and terminated at either 10 min or 3 h post-injection (n = 3 per group; a total of 18 mice). Following termination with a lethal intraperitoneal injection (200 μL pentobarbital), mice were tracheotomised and lungs were inflated with pre-chilled 4% PFA via a tracheal tube. Subsequently, the lungs were excised and post-fixed in 4% PFA for 24 h at 4 °C, followed by 48 h incubation in 30% sucrose at 4 °C. Lungs were snap-frozen, cryosectioned at a thickness of 10 µm, and stored at −20 °C until immunohistochemical staining for VCAM-1.

Frozen sections were thawed for 20 min at room temperature (RT), quenched with 1% *v*/*v* of an aqueous solution of hydrogen peroxide (30% *w*/*v*) in methanol (Sigma Aldrich, Saint Louis, MO, USA), and blocked with 1% normal horse serum in PBS (Vector Laboratories, Newark, CA, USA) for 1 h. Sections were incubated with primary rat anti-VCAM-1 antibody (M/K 2 clone, 1/500; Cambridge Bioscience, Cambridge, UK) at 4 °C overnight, washed with PBS containing 0.01% Tween-20 (Sigma Aldrich) and incubated with secondary horse anti-rat biotinylated antibody (1/100; BA-4001, Vector Laboratories). Slides were washed and incubated in a Vectorstain Elite ABC kit (1:100 per substrate; Vector Laboratories) for 45 min. Peroxidase was visualised using 3,3′-diaminobenzidine (DAB; Sigma Aldrich). Sections were counterstained with hematoxylin (Sigma Aldrich) for nucleus definition.

Representative sections were chosen across the lung sample (15 sections; n = 5); 15 metastatic colonies per lung sample were chosen, and fixed-size fields of view (FOVs; 460 × 460 μm) were marked as having metastatic colonies at the centre of the FOV. Within the FOVs, metastases and vessels were manually segmented, and the number of adherent MPIO on or within micrometastasis-associated pulmonary capillaries was counted using bright-field microscopy. From these data, both the number of MPIO per tumour area and the number of MPIO per VCAM-1 positive vessels were calculated. The level of non-specific retention was determined from the equivalent FOVs in naïve animals injected with VCAM-MPIO or IgG-MPIO. In order to compare the non-specific retention in naïve animals with the non-specific retention in metastasis-bearing animals, a further 15 FOV per metastasis-bearing lung were chosen from metastasis-free areas on each section and were also assessed for MPIO binding.

### 4.3. Radiolabelled VCAM-, IgG- or BSA-MPIO Conjugates

#### 4.3.1. [^89^Zr]-DFO-VCAM-MPIO and [^89^Zr]-DFO-IgG-MPIO Synthesis

To allow radiolabelling with ^89^Zr, rat anti-mouse VCAM-1 antibody (M/K 2 clone, Cambridge Bioscience, Cambridge, UK) or rat IgG-1 isotype control antibody (Cambridge Bioscience) was prepared in sodium bicarbonate buffer (0.1 M, pH 8.9), and incubated with the bifunctional chelator, 1-(4-isothiocyanatophenyl)-3-[6,17-dihydroxy-7,10,18,21- tetraoxo-27-(N-acetylhydroxylamino)-6,11,17,22-tetraazaheptaeicosine] thiourea (p-SCN-Bn-desferrioxamine, p-SCN-Bn-DFO; Macrocyclics, Dallas, TX, USA), at 10-fold molar excess, for 1 h at 37 °C. The excess unreacted chelator was removed by size-exclusion chromatography (G50 Sephadex, Global Life Sciences Solutions Operations UK Ltd., Little Chalfont, UK) [46]. The product reagents DFO-VCAM and DFO-IgG were then conjugated to MPIO as described above. Subsequently, ^89^Zr^4+^ (dissolved in 1 M oxalic acid; Perkin Elmer, Waltham, MA, USA) was rendered to neutral pH using 1 M Na_2_CO_3_. DFO-VCAM-MPIO or DFO-IgG-MPIO were prepared in phosphate-buffered saline (pH 7.4) and reacted with the appropriate amount of ^89^Zr for 1 h at RT, resulting in ^89^Zr-DFO-VCAM-MPIO or ^89^Zr-DFO-IgG-MPIO. The conjugates were subsequently washed twice with 1 mL PBS and pelleted on a Dynal magnetic separator (Thermo Fisher Scientific, Waltham, MA, USA) for 2 min to remove unreacted ^89^Zr. Radiolabelling yields for ^89^Zr-DFO-VCAM-MPIO and ^89^Zr-DFO-IgG-MPIO after purification were 10.6 ± 5.5% (n = 4) and 8.91 ± 6.6% (n = 4), respectively. The radiochemical purity of each radiotracer was determined by magnetic separation and was routinely >98 %.

#### 4.3.2. In Vitro Assessment of [^89^Zr]-DFO-VCAM-MPIO Binding to VCAM-1

To confirm the binding affinity of [^89^Zr]-DFO-VCAM-MPIO and [^89^Zr]-DFO-IgG-MPIO to VCAM-1, radioimmunoassays were performed in triplicate with either immobilised recombinant mouse VCAM-1 (rmVCAM-1) or TNF-stimulated human umbilical vein cells (HUVECs), as described previously [47].

To assess binding to rmVCAM-1, 96-well flat-bottom plates (Cellstar, Sigma Aldrich, Saint Louis, MO, USA) were coated with rmVCAM-1 (seeding concentration = 0.5 μg/mL; Sino Biological Inc., Beijing, China) and were incubated with 1 nM of [^89^Zr]-DFO-VCAM-MPIO, [^89^Zr]-DFO-IgG-MPIO, [^89^Zr]-DFO-VCAM or [^89^Zr]-DFO for 30 min at 4 °C with or without VCAM-1 antibody competition (100 nM; M/K 2 clone). The plates were washed, and radioactivity in each well was determined by quantitative autoradiography. To assess the binding of VCAM-1 to endothelial cells, HUVECs were seeded onto 96-well flat-bottom plates (5 × 10^3^ per well) and grown to 80% confluence. To stimulate VCAM-1 expression, 2/3 of the wells were incubated with 10 ng/mL human TNF (Peprotech, London, UK) for 18 h. Unstimulated cells were used as controls. Subsequently, 1 nM of [^89^Zr]-DFO-VCAM-MPIO, [^89^Zr]-DFO-IgG-MPIO, [^89^Zr]-DFO-VCAM, and [^89^Zr]-DFO were added to the wells and the plates were incubated for 30 min at 4 °C. Subsequently, the experiment was repeated with radioactive agents incubated in competition with a VCAM-1 antibody (100 nM; M/K 2 clone). The plates were washed, and the radioactivity in each well was determined by quantitative autoradiography. All experiments were carried out in triplicates.

### 4.4. In Vivo Imaging

#### 4.4.1. In Vivo Assessment of [^89^Zr]-DFO-VCAM-MPIO Binding in Pulmonary Metastasis Model

A total of 16 mice were anaesthetised with 3% isofluorane (IsoFlo) in room air, and subsequently, anaesthesia was maintained with 1–2% isoflurane until termination. Metastasis-bearing mice were injected intravenously via a tail vein with either [^89^Zr]-DFO-VCAM-MPIO or [^89^Zr]-DFO-IgG-MPIO: 1.6 mg Fe/kg in 100 μL sterile PBS; 5 MBq activity per injection (n = 4 per group). As further controls, naïve mice were injected with either [^89^Zr]-DFO-VCAM-MPIO or [^89^Zr]-DFO-VCAM (n = 4 per group) as described above.

PET imaging was performed using the Inveon PET/CT system (Siemens Preclinical Solutions, Hague, The Netherlands). Circulatory half-life measurements indicated rapid clearance from the blood pool (half-life 39.9 s; Appendix A). Consequently, animals underwent PET imaging immediately after contrast agent injection. Throughout each imaging session, the mice were maintained at 37 °C, and the respiration rate was monitored (60–100 respirations/min). Mice were placed prone and head first in the imaging cradle, and a cannula was inserted into the lateral tail vein. CT-based attenuation correction (continuous rotation, 220 degrees, 120 steps, 65 kV, 500 μA, 200 ms exposure time, 3072 × 2048 matrix, binning of 4, 3 bed positions with 28.45% overlap) was performed before each PET emission scan. The CT acquisition was also used for anatomical referencing. For all compounds, a 1 h whole-body dynamic PET acquisition was performed. Following the CT scan, PET emission data acquisition was initiated (3600 s, ^89^Zr, 3.432 ns timing window, 511 keV photopeak energy level, 350–650 keV energy level discrimination) and the radiotracer was injected via the tail vein cannula. Data were binned in 24 time frames (15 × 20 s, 7 × 300 s, and 2 × 600 s) with a span of 3 and a ring difference of 79. Global dead-time correction and Fourier re-binning were applied to all datasets. Dynamic histograms were reconstructed using a 2-dimensional filtered back-projection (2D-FBP) algorithm, a Ramp projection filter, a 0.5/mm Nyquist projection cut-off value, no zoom, and a matrix size of 128 × 128 × 159 (sagittal × coronal × transversal). Both histogramming and image reconstruction were conducted with the Inveon Acquisition Workplace software (IAW, version 1.5). Image analysis was performed using the Inveon Research Workplace software (IRW, version 4.2).

#### 4.4.2. Post Mortem Assessment of [^89^Zr]-DFO-VCAM-MPIO Binding in Pulmonary Metastasis Model

Two metastasis-bearing mice injected with either [^89^Zr]-DFO-VCAM-MPIO or [^89^Zr]-DFO-IgG-MPIO were sacrificed 10 min post-administration and placed inside 50 mL Falcon tubes. The tubes were imaged over a 2 h time frame with PET using the same parameters as for in vivo imaging. After the end of the scan, the mouse injected with [^89^Zr]-DFO-VCAM-MPIO was kept at 4 °C for 24 h to reduce radioactivity, and then all organs below the diaphragm were surgically removed, and the upper body was placed inside a 50 mL Falcon tube. Subsequently, the mouse was rescanned for 60 h using the same PET parameters as those used for in vivo imaging.

### 4.5. Ex Vivo Analysis

#### 4.5.1. Ex Vivo Assessment of MPIO Biodistribution

A total of 16 metastasis-bearing mice or 16 naïve animals were injected intravenously via a lateral tail vein with 1.6 mg of Fe/kg of body weight of either [^89^Zr]-DFO-VCAM-MPIO or [^89^Zr]-DFO-IgG-MPIO (0.5 MBq per injection) and terminated with lethal injection (200 μL pentobarbital) 10 min later (n = 8 per group). Blood samples were collected from each animal immediately after incision into the diaphragm by cutting the inferior vena cava and allowing the blood to pool within the diaphragm-constrained area. Subsequently, the heart, lungs, liver, spleen, stomach, large intestine, small intestine, pancreas, kidneys, and samples of bone (right ulna), skin (abdomen), and visceral fat (abdomen) were excised, washed in PBS (to eliminate blood), and left to dry on high-absorbency paper. Selected organs, tissues, and blood were placed in 5 mL tubes of known weight (Thermo Fisher Scientific, Waltham, MA, USA), and the tubes were placed in a gamma counter (1470 WIZARD gamma counter, Perkin Elmer, Waltham, MA, USA) to determine radioactivity per sample. Tubes containing the organs were weighed to calculate the mass of the organ, and subsequently, the percentage of injected dose per g (%ID/g) of the organ was calculated.

#### 4.5.2. Ex Vivo Autoradiography of MPIO Accumulation in Lungs

Groups equivalent to those described for the biodistribution study were used for ex vivo autoradiography (n = 2 per group; 8 mice in total). Lungs were excised and treated as described above. The lungs were kept in sucrose for 48 h and then cryosectioned as described above. After cryosectioning, the mounted lung sections were placed on a Cylcone^®^ storage phosphor screen and exposed for 12 or 24 h at room temperature. Subsequently, the same lung sections were stained immunohistochemically for VCAM-1, as described above.

## 5. Conclusions

In conclusion, we present here a novel imaging method for the detection of pulmonary micrometastases in mice by targeting activated pulmonary endothelium using a PET-detectable molecularly targeted contrast agent, [^89^Zr]-DFO-VCAM-MPIO. By keeping the microparticle platform used for our molecular MRI contrast agents, we both retain their clearance characteristics, yielding highly specific target signal changes and providing a dual-modality agent that can be used to detect micrometastases in different tissue beds with the optimal imaging approach for each. The agent is in itself a platform, as the basic MPIO component can be conjugated to a range of different targeting and contrast molecules to create varied and multifunctional agents. Looking ahead, the next steps for research should focus on further refining the probe’s specificity and efficiency, leading to clinical trials. Downstream, exploring the integration of this imaging technology into routine clinical practice will be crucial. At the research level, expanding the application of this probe to other types of cancer could provide comprehensive insights into its versatility and effectiveness across different oncological contexts. These steps will be pivotal in moving from experimental application to a standard component of oncological imaging and treatment planning, paving the way for advances in personalised medicine and improved prognosis in patients with cancer.

## Figures and Tables

**Figure 1 ijms-25-07160-f001:**
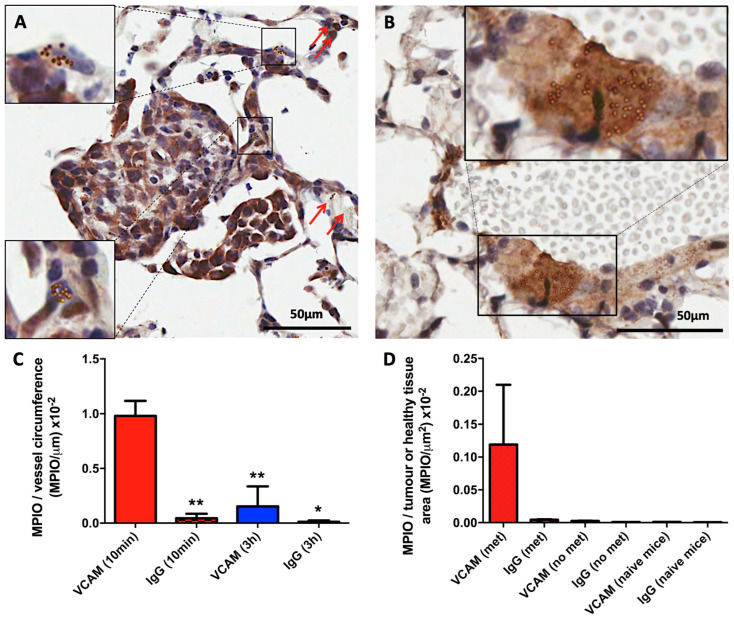
Quantitative and spatial analysis of VCAM-MPIO and IgG-MPIO binding in lung tissues of in metastasis-bearing and naïve mice. (**A**,**B**) Representative images of lung tissue acquired from the metastatic mouse model (day 10), 10 min post-intravenous injection of VCAM-MPIO, stained for VCAM-1 (brown) with a nuclear counterstain (blue). (**A**) Substantial numbers of MPIO are associated with the metastasis microvasculature (red arrows and higher magnification areas), and also on nearby larger vessels showing high VCAM-1 expression (**B**). (**C**,**D**) Immunohistochemical quantitation of MPIO binding in metastasis-bearing or naïve lungs (10 min or 3 h after i.v. injection). (**C**) Bar graphs represent the number of MPIO per μm vessel circumference and (**D**) the number of MPIO per μm^2^ in areas of metastasis (met), areas of no apparent metastasis in the metastasis model (no met), or healthy lung tissue (naïve mice) 10 min after i.v. injection. At 10 min post-administration VCAM-MPIO showed significantly greater retention on VCAM-1 expressing vessels associated with micrometastases compared to IgG-MPIO control). Similar levels of targeted and control agent retention at 3 h indicate rapid clearance of MPIO from the tissue. * *p* < 0.01; ** *p* < 0.005 compared to VCAM-MPIO retention at 10 min post-injection. Non-specific retention was negligible compared to specific binding.

**Figure 2 ijms-25-07160-f002:**
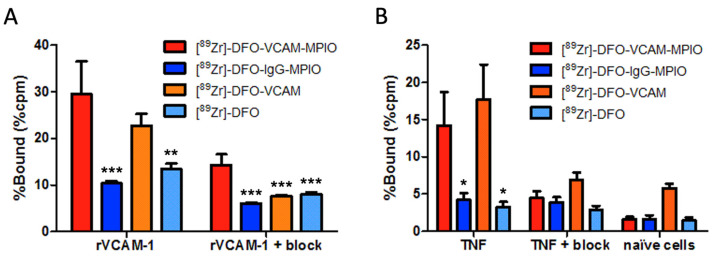
Specificity and competitive inhibition of [^89^Zr]-DFO-VCAM-MPIO binding to VCAM-1 and TNF-stimulated HUVECs: radioimmunoassay evaluation. (**A**) Radioimmunoassay plots showing the % of bound radiolabelled agents against recombinant VCAM-1 or (**B**) TNF-stimulated HUVECs with 30 min static incubation, +/− competition with VCAM-1 antibody (blocking). (**A**) [^89^Zr]-DFO-VCAM-MPIO showed significantly greater % binding compared to both negative controls [^89^Zr]-DFO-IgG-MPIO and [^89^Zr]-DFO (*p* < 0.005). Following blockade with VCAM-1 antibody, [^89^Zr]-DFO-VCAM-MPIO binding was significantly reduced (*p* < 0.001). (**B**) [^89^Zr]-DFO-VCAM-MPIO showed significantly greater % binding to TNF-stimulated HUVECs compared to both negative controls [^89^Zr]-DFO-IgG-MPIO and [^89^Zr]-DFO (*p* < 0.001). Binding was substantially reduced with the inclusion of the blocking antibody and was negligible on unstimulated cells. One-way ANOVA with Tukey post-hoc tests; * *p* < 0.01; ** *p* < 0.005; *** *p* < 0.001; all comparisons are against [^89^Zr]-DFO-VCAM-MPIO.

**Figure 3 ijms-25-07160-f003:**
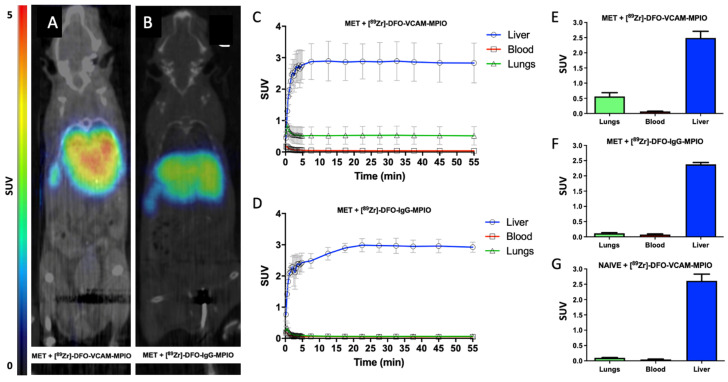
Differential uptake of radiolabelled VCAM-MPIO and IgG-MPIO in metastasis-bearing and naïve mice. (**A**,**B**) In vivo, non-gated maximum intensity plots obtained from 3D PET datasets from mice with pulmonary metastases and injected with (**A**) [^89^Zr]-DFO-VCAM-MPIO or (**B**) [^89^Zr]-DFO-IgG-MPIO. (**A**) Marked radioactivity is evident in the lungs, liver, and spleen in the metastasis-bearing mouse injected with [^89^Zr]-DFO-VCAM-MPIO, but (**B**) only from the liver and spleen in the metastasis-bearing mouse injected with the isotype control agent [^89^Zr]-DFO-IgG-MPIO. (**C**,**D**) Representative average activity uptake curves for (**C**) [^89^Zr]-DFO-VCAM-MPIO and (**D**) [^89^Zr]-DFO-IgG-MPIO in liver, blood, and lungs from metastasis-bearing mice (n = 4 per group). The liver shows rapid and high accumulation of radioactivity in both targeted and control groups. (**C**) Lungs showed a higher accumulation of activity for the [^89^Zr]-DFO-VCAM-MPIO injected mice than (**D**) the [^89^Zr]-DFO-IgG-MPIO group. (**E**–**G**) Graphs showing average activity uptake over a duration of 55 min for liver, blood and lung from metastasis-bearing mice injected with (**E**) [^89^Zr]-DFO-VCAM-MPIO or (**F**) [^89^Zr]-DFO-IgG-MPIO, or (**G**) from naïve mice injected with [^89^Zr]-DFO-VCAM-MPIO. In all cases, the liver showed significantly higher uptake than other tissues (*p* < 0.0001), reflecting the role of the liver in scavenging unbound MPIO from the blood. (**E**) Lungs from metastasis-bearing mice injected with the targeted agent showed significantly (*p* < 0.0005) higher uptake than (**F**) lungs from mice injected with the control agent or (**G**) lungs from naïve mice injected with the targeted agent (*p* < 0.0005).

**Figure 4 ijms-25-07160-f004:**
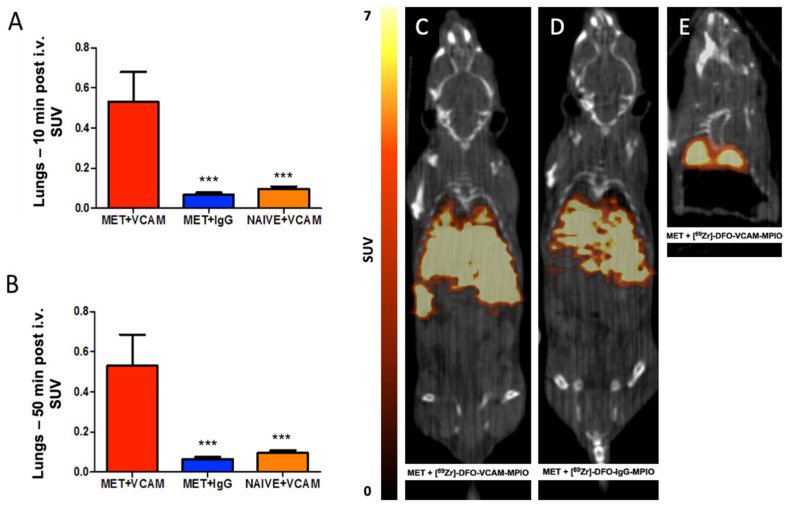
Temporal and spatial analysis of radiolabelled VCAM-MPIO or IgG-MPIO uptake in metastasis-bearing and naïve mice. (**A**,**B**) In vivo average activity uptake in the lungs of mice injected with (**A**) [^89^Zr]-DFO-VCAM-MPIO at 10 and (**B**) 50 min post-administration (n = 4 per group) showed significantly greater retention in metastasis-bearing lungs compared to naïve lungs or to metastasis-bearing mice injected with [^89^Zr]-DFO-IgG-MPIO (*** *p* < 0.0001). (**C,D**) Post-mortem (10 min post i.v.) 2 h acquisition whole-body PET images of mice injected with (**C**) [^89^Zr]-DFO-VCAM-MPIO or (**D**) [^89^Zr]-DFO-IgG-MPIO showing marked radioactivity in the liver and spleen. (**C**) Lungs from the mouse injected with [^89^Zr]-DFO-VCAM-MPIO also show notable radioactivity (**D**), whereas signals from the lungs of the mouse injected with [^89^Zr]-DFO-IgG-MPIO do not. (**E**) The mouse injected with [^89^Zr]-DFO-VCAM-MPIO in image (**C**) was further examined using a long PET acquisition (60 h) after removing the body and organs below the diaphragm, to confirm that the apparent lung signal, as seen in image (**C**), originated from the lungs and was not a partial volume effect from liver and spleen.

**Figure 5 ijms-25-07160-f005:**
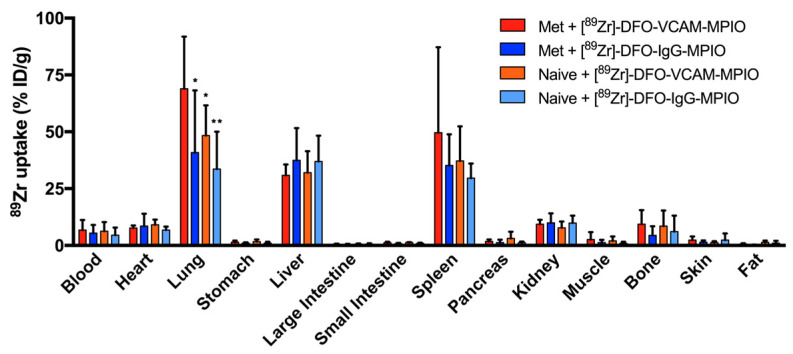
Organ-specific biodistribution of radiolabelled VCAM-MPIO or IgG-MPIO in metastasis-bearing and naïve mice. Bar graph showing the average % of injected dose per mass of organ for all groups (n = 8 per group). In the lungs, metastasis-bearing mice injected with [^89^Zr]-DFO-VCAM-MPIO (red) showed significantly (*p* < 0.05) greater radiation yield than either the control [^89^Zr]-DFO-IgG-MPIO injected groups (blue and cyan) or naïve mice injected with [^89^Zr]-DFO-VCAM-MPIO (orange). The liver and spleen showed the next highest levels of radioactivity uptake, supporting their role in the scavenging of non-bound particles in circulation. The kidney and bone showed comparable levels of %ID/g to the blood and heart, and thus, they are likely to be background circulating levels. One-way ANOVA; * *p* < 0.05; ** *p* < 0.01.

**Figure 6 ijms-25-07160-f006:**
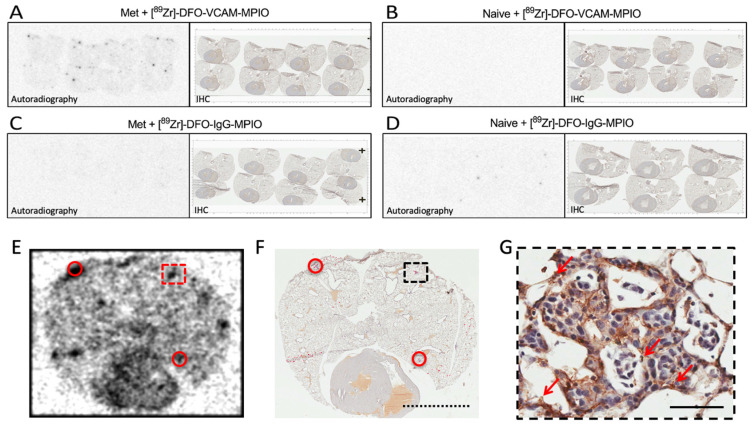
Comparative autoradiographic and immunohistochemical analyses and colocalisation of foci with radiolabelled VCAM-MPIO and micrometastases. (**A**–**D**) Paired images of lung tissue autoradiographic data (left; 10 h exposure) and bright-field micrographs of the associated lung sections (right) stained immunohistochemically for VCAM-1 expression (brown) for (**A**) metastasis-bearing or (**B**) naïve lungs treated with [^89^Zr]-DFO-VCAM-MPIO; (**C**) metastasis-bearing or (**D**) naïve lungs treated with [^89^Zr]-DFO-IgG-MPIO. (**A**) Metastasis-bearing lungs + [^89^Zr]-DFO-VCAM-MPIO showed sparse basal activity in all areas of the lung sections and localised intense signals (dark spots). (**B**–**D**) All other groups showed no or minimal activity. (**E**,**F**) Autoradiographic images from the lung tissue section after 24 h exposure and immunohistochemical images from the same lung section manually inspected under magnification. Foci of high activity on the autoradiographic plate (e.g., red dashed square and red circles) colocalise with VCAM-1 expressing micrometastatic areas (black dashed square and red circles). The high activity foci on the autoradiographic plate that do not correlate with apparent metastases do correlate with areas where MPIO were manually scored ([**F**]; hard to distinguish small red marks). Dotted scale bar = 5 mm. (**G**) Black dashed square in (**F**) is shown at a higher resolution; arrows indicate the presence of MPIO. Nuclei are stained blue. Scale bar = 50 μm.

## Data Availability

The datasets supporting the findings of the current study are available from the corresponding author upon request.

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
