# Peer review of "In Vivo PET Detection of Lung Micrometastasis in Mice by Targeting Endothelial VCAM-1 Using a Dual-Contrast PET/MRI Probe"

_ijms, 2024, doi:10.3390/ijms25137160_

Round 1
Reviewer 1 Report
Comments and Suggestions for Authors
The manuscript developed a new contrast agent for PET/MRI imaging. This is a very interesting study. The author tested this contrast agent on metastases bearing and naive mice. This could be a very useful tool to detect and study the lung and brain micrometastases.
-
In Figure 3 legend, it says in met and naive mice, but based on the description, both A and B are met mice. And the (A-B) in line 228 is a bit confusing.
-
Some greek letters in the text are not shown correctly, i.e. line 23, 30, 142.
-
The results presented in figure 5 are a bit confusing. Why does the lung have a quite high update of Zr for the met+Zr-IgG and the naive+Zr groups? Doesn’t this imply that lung updates Zr even for healthy mice? But in the results shown on figure 3 C-G and figure 4 A-B, the SUV curve does not show any activity for met+Zr-IgG and the naive+Zr mice.
Author Response
- In Figure 3 legend, it says in met and naive mice, but based on the description, both A and B are met mice. And the (A-B) in line 228 is a bit confusing.
We thank the reviewer for noticing this error. (A-B) was misplaced and should have been at the beginning of the legend. Graph G shows data from naïve mice, so we have kept the figure title the same, but the legend has now been corrected and reads as follows:
Line 247-251:
Figure 3. Differential uptake of radiolabeled VCAM-MPIO and IgG-MPIO in metastasis-bearing and naive and naïve mice
(A-B) In vivo, non-gated maximum intensity plots obtained from 3D PET datasets from mice with pulmonary metastases and injected with (A) [89Zr]-DFO-VCAM-MPIO or (B) [89Zr]-DFO-IgG-MPIO.
- Some greek letters in the text are not shown correctly, i.e. line 23, 30, 142.
Corrected.
- The results presented in figure 5 are a bit confusing. Why does the lung have a quite high update of Zr for the met+Zr-IgG and the naive+Zr groups? Doesn’t this imply that lung updates Zr even for healthy mice? But in the results shown on Figure 3 C-G and Figure 4 A-B, the SUV curve does not show any activity for met+Zr-IgG and the naive+Zr mice.
The difference in the observed uptake values of Zirconium (Zr) in the lung tissue across these figures can be attributed to the distinct measurement methodologies used. In Figure 5, we present the data as the percentage of injected dose per gram of tissue (%ID/g; injected dose Bq / Tissue Weight g), which shows a relatively higher uptake in the lungs owing to the lung's low mass. This normalization makes the measured activity appear elevated when compared to other tissues. Conversely, the data presented in Figures 3 and 4 use the Standardized Uptake Value (SUV), which considers the radioactivity concentration in the tissue normalized over the body weight-adjusted total injected dose, Activity concentration in the tissue (Bq/mL) / Injected dose normalized over body weight (Bq/g).
The SUV is a more diluted measurement and often does not highlight uptake in low mass organs like the lungs, unless there is significant specific uptake or retention of the tracer. The proportional ratios between groups across these figures are consistent, and the discrepancies arise from the normalization methods employed in each type of measurement. Thus, %ID/g and SUV provide different perspectives on tracer distribution, with %ID/g potentially emphasizing apparent uptake in smaller/low mass organs, whilst SUV offers a whole-body normalized view that may downplay these effects.

Reviewer 2 Report
Comments and Suggestions for Authors
This manuscript presents a significant advancement in the early detection of lung micrometastases using a novel dual-contrast PET/MRI probe targeting VCAM-1. Overall, the manuscript is well-written and presents significant findings. With the suggested improvements, particularly in organization and clarity, it will be more accessible and impactful for readers.
The order of the presentation should be as follows;
-
Abstract
-
Materials and Methods
-
Result
-
Discussion
-
Conclusion
Other comments:
-
Abstract:
-
Include potential clinical implications..
-
Also, add a sentence on the potential impact of the findings on early cancer detection and patient outcomes.
-
Please re-check the unit of measurement mentioned across the manuscript. I believe you meant to mention “Micrometer (µm)” especially in the abstract.
-
Introduction:
-
Discuss the limitations of CT and 18F-FDG PET in detecting micrometastases.
-
Explain why VCAM-1 is a promising target for early detection.
-
State the objective of developing a dual-contrast PET/MRI probe.
-
Material and Methods:
-
Detailed but lacks structure. Readers can easily get lost reading the Materials and Methods section of the manuscript. Use subheadings for better organization.
-
The following subheadings can be useful: “Synthesis and Loading of Conjugates”, “Animal Models”, “In Vivo Imaging”, “Ex Vivo Analysis”, and “Statistical Analysis”.
-
Result:
-
Ensure a logical flow from synthesis to in vivo and ex vivo results.
-
Discussion:
-
Suggest specific future research directions.
-
The discussion effectively interprets the results but can be more concise
-
Conclusion:
-
Highlight the broader implications of the findings.
-
Mention the next steps for research or clinical application.
-
References
-
References older than 20 years ago are only ideal incases of history, evolution etc.
Author Response
The order of the presentation should be as follows;
- Abstract
- Materials and Methods
- Result
- Discussion
- Conclusion
We thank the reviewer for this comment. We also believe that the format suggested might provide a better flow, with the Material and Methods presented after the introduction. However, this format conflicts with the journal’s template, which places the Material and Methods after the Conclusions. Therefore, we do not know if this change is allowed from an editorial perspective. We are happy to change the order if the Editors agree.
Other comments:
- Abstract:
- Include potential clinical implications..
- Also, add a sentence on the potential impact of the findings on early cancer detection and patient outcomes.
We agree that these additions would enhance the reader's understanding of the significance of our research, and we do already note that earlier detection could dramatically improve treatment. However, we face a challenge owing to the strict word limit of 200 words imposed by the journal. Our study's complexity and the need to include critical methodological details make it particularly difficult to expand on clinical applications without omitting essential information required for the understanding of our study's methodology and core findings. We have tried to balance the inclusion of fundamental data with broader implications within the allowed word limit. We would appreciate any further guidance on how we might better manage this balance or if there is specific information that could be condensed or omitted to accommodate these valuable additions.
- Please re-check the unit of measurement mentioned across the manuscript. I believe you meant to mention “Micrometer (µm)” especially in the abstract.
This is correct. There was a font compatibility issue that is now corrected.
- Introduction:
- Discuss the limitations of CT and 18F-FDG PET in detecting micrometastases.
We have added some discussion of CT and 18F-FDG PET in detecting micrometastases in the ‘Introduction’ section, which now reads:
Line 54-69: 18F-FDG PET complements the CT resolution, offering information about uncertain small lesions and their metabolic activity. It is important to note, however, that the detection sensitivity of 18F-FDG PET relies on an accumulation of highly metabolically active cells (both proliferating tumour cells and inflammatory cells) to allow tumour detection/diagnosis, which typically reflects a relatively late stage of tumour development and leaves micrometastases undetected [7,8]. This resolution limitation results in decreased sensitivity for low-volume disease, leading to potential false negatives [9,10]. Moreover, CT scans depend on size and density contrasts to identify abnormalities, and such contrasts may not be evident in early-stage disease, causing micrometastases to be over-looked [11]. At the same time, 18F-FDG is not exclusive to tumors and accumulates in any glucose-avid tissues, including those affected by inflammation. This non-specific uptake can obscure the presence of disease or mimic it, complicating the differentiation between cancerous and non-cancerous conditions [12,13]. Owing to the lack of imaging sensitivity for the early stages of tumour cell extravasation from the circulation and micrometastatic formation, clinical therapy is often applied at a time that is beyond the point of effective intervention.
- Akin, O.; Brennan, S.B.; Dershaw, D.D.; Ginsberg, M.S.; Gollub, M.J.; Schöder, H.; Panicek, D.M.; Hricak, H. Advances in Oncologic Imaging. CA: A Cancer Journal for Clinicians 2012, 62, 364–393, doi:10.3322/caac.21156.
- Shen, K.; Liu, B.; Zhou, X.; Ji, Y.; Chen, L.; Wang, Q.; Xue, W. The Evolving Role of 18F-FDG PET/CT in Diagnosis and Prognosis Prediction in Progressive Prostate Cancer. Front. Oncol. 2021, 11, doi:10.3389/fonc.2021.683793.
- Ben-Haim, S.; Ell, P. 18F-FDG PET and PET/CT in the Evaluation of Cancer Treatment Response. Journal of Nuclear Medicine 2009, 50, 88–99, doi:10.2967/jnumed.108.054205.
- Rampinelli, C.; Calloni, S.F.; Minotti, M.; Bellomi, M. Spectrum of Early Lung Cancer Presentation in Low-Dose Screening CT: A Pictorial Review. Insights Imaging 2016, 7, 449–459, doi:10.1007/s13244-016-0487-4.
- Bunyaviroch, T.; Coleman, R.E. PET Evaluation of Lung Cancer. Journal of Nuclear Medicine 2006, 47, 451–469.
- Abouzied, M.M.; Crawford, E.S.; Nabi, H.A. 18F-FDG Imaging: Pitfalls and Artifacts. Journal of Nuclear Medicine Technology 2005, 33, 145–155.
- Explain why VCAM-1 is a promising target for early detection.
We have clarified why VCAM-1 is promising target for early detection in paragraph 2 of the ‘Introduction’ section, which now reads:
Line 72-91: VCAM-1 We have previously demonstrated that it is possible to detect micrometastases in mouse brain very early in their development, using molecularly-targeted magnetic resonance imaging (MRI) via the associated endothelial activation and upregulation of endovascular adhesion molecules [14–16]. These endothelial adhesion molecules are rapidly upregulated in response to disease or injury, and mediate leukocyte rolling, adhesion and transmigration across the vascular wall [17]. There is now evidence to suggest that tumor cells also use such adhesion molecules to promote their adhesion to and extravasation across the vascular endothelium [18]. One such adhesion molecule is vascular cell adhesion molecule 1 (VCAM-1), which is upregulated very early in brain metastasis formation [19] and has also been shown to enhance endothelial cell recruitment for new blood vessel formation, thus supporting tumor growth and spread [20]. The early and marked upregulation of VCAM-1 marks this molecule as a potential early biomarker for detecting metastatic activity, offering advantages over traditional imaging markers that may not clearly distinguish between tumor stages or identify early metastatic spread. To this end, using an MRI contrast agent based on microparticles of iron oxide (MPIO) and targeted to VCAM-1, we have previously demonstrated that VCAM-1 provides a sensitive and specific endothelial biomarker of early brain metastasis [14,15], as well as improving detection of the tumour-brain interface [21].
Previous studies have also shown that VCAM-1 is markedly upregulated in both lung and liver metastases [22,23], with little expression on normal lung endothelium.
- Zhang, D.; Bi, J.; Liang, Q.; Wang, S.; Zhang, L.; Han, F.; Li, S.; Qiu, B.; Fan, X.; Chen, W.; et al. VCAM1 Promotes Tumor Cell Invasion and Metastasis by Inducing EMT and Transendothelial Migration in Colorectal Cancer. Front. Oncol. 2020, 10, doi:10.3389/fonc.2020.01066.
- Khatib, A.-M.; Auguste, P.; Fallavollita, L.; Wang, N.; Samani, A.; Kontogiannea, M.; Meterissian, S.; Brodt, P. Characteriza-tion of the Host Proinflammatory Response to Tumor Cells during the Initial Stages of Liver Metastasis. Am J Pathol 2005, 167, 749–759, doi:10.1016/S0002-9440(10)62048-2.
- State the objective of developing a dual-contrast PET/MRI probe.
We have added a paragraph at the end of the introduction that explicitly states the objective of developing a dual-contrast PET/MRI probe, which reads:
Line 113-117: The aim of this study, therefore, was to develop a dual-contrast PET/MRI probe targeting VCAM-1 to enable early detection of micrometastases in both brain and lung using a single contrast agent. This dual-modality approach not only promises to improve the accuracy of metastasis detection, but may also facilitate the early initiation of targeted therapies, potentially improving patient outcomes significantly.
- Material and Methods:
- Detailed but lacks structure. Readers can easily get lost reading the Materials and Methods section of the manuscript. Use subheadings for better organization.
- The following subheadings can be useful: “Synthesis and Loading of Conjugates”, “Animal Models”, “In Vivo Imaging”, “Ex Vivo Analysis”, and “Statistical Analysis”.
We appreciate the reviewer's recommendations for improving the structure of the Materials and Methods section. We recognize the value of using subheadings to enhance clarity and guide the reader through the methodology more effectively. However, we would like to address the specific suggestion of grouping all of the "Synthesis and Loading of Conjugates" and "Ex Vivo Analysis" into single sections, respectively.
In our study, the initial sections cover the synthesis, loading, and ex vivo binding of non-radiolabeled VCAM-, IgG-, or BSA-MPIO conjugates. We believe it is crucial for readers to fully understand these aspects before progressing to the sections concerning radiolabeled conjugates. Grouping the non-radiolabeled and radiolabeled conjugates separately clarifies these distinct phases of our research and aids in comprehending the progression from one stage to the next.
Therefore, whilst we agree with the general suggestion of introducing subheadings for better navigation, we propose a slightly modified organization to maintain the logical flow and distinction between the non-radiolabeled and radiolabeled components of the study. We have modified the ‘Materials and Methods’ section with the following subheadings and sub subheadings order:
- Materials and Methods
- Pulmonary Metastasis Mouse Model
- Non-Radiolabelled VCAM-, IgG- or BSA-MPIO Conjugates
- Synthesis of Conjugates
- Assessment of Antibody Loading
- Ex Vivo Assessment of VCAM-MPIO Binding in Pulmonary Metastasis Model
- Radiolabelled VCAM-, IgG- or BSA-MPIO Conjugates
- [89Zr]-DFO-VCAM-MPIO and [89Zr]-DFO-IgG-MPIO Synthesis
- In Vitro Assessment of [89Zr]-DFO-VCAM-MPIO Binding to VCAM-1
- In vivo Imaging
- In Vivo Assessment of [89Zr]-DFO-VCAM-MPIO Binding in Pulmonary Metastasis Model
- Post-Mortem Assessment of [89Zr]-DFO-VCAM-MPIO Binding in Pulmonary Metastasis Model
- Ex Vivo Analysis
- Ex Vivo Assessment of MPIO Biodistribution
- Ex Vivo Autoradiography of MPIO Accumulation in Lungs
- Result:
- Ensure a logical flow from synthesis to in vivo and ex vivo results.
Based on the organization outlined in the Materials and Methods, we have structured the ‘Results’ section with the following headings and subheadings:
- Results
- Non-Radiolabelled VCAM-, IgG- or BSA-MPIO Conjugates
- Synthesis and Loading of Conjugates
- Ex Vivo Quantitation of VCAM-MPIO Binding in Pulmonary Metastasis Model
- Radiolabelled VCAM-, IgG- or BSA-MPIO Conjugates
- In Vitro Binding of [89Zr]-DFO-VCAM-MPIO
- In Vivo Imaging
- In Vivo PET Imaging of [89Zr]-DFO-VCAM-MPIO in Pulmonary Metastasis Model
- Post-mortem PET Imaging of [89Zr]-DFO-VCAM-MPIO in Pulmonary Metastasis Model
- Ex Vivo Analysis
- Ex Vivo Biodistribution of [89Zr]-DFO-VCAM-MPIO
- Lung Autoradiography of [89Zr]-DFO-VCAM-MPIO Accumulation in Lungs
- Non-Radiolabelled VCAM-, IgG- or BSA-MPIO Conjugates
- Discussion:
- Suggest specific future research directions.
We have included the following paragraph in the Discussion in response to this suggestion:
Line 512-524: In considering the future trajectory of imaging technology, the development of an [18F]-labeled VCAM-MPIO variant would leverage the superior imaging qualities of 18F compared to 89Zr, including enhanced spatial resolution as a result of its lower positron range and a reduced radioactivity exposure for patients, owing to its shorter half-life. At the same time, implementing respiratory and cardiac gating would improve sensitivity, potentially enabling individual radiotracer hot spots to be correlated with micrometastases on the PET scans. Downstream, co-targeting of additional biomarkers alongside VCAM-1 has the potential to provide a more nuanced view of tumor biology and the microenvironment. Finally, integrating machine learning algorithms with dual-modality imaging data presents an opportunity to refine diagnostic accuracy. Training these algorithms on extensive datasets that meld PET and MRI outputs could foster predictive models that more accurately forecast disease progression and therapeutic responses leading to a new level of precision in diagnostic imaging.
- The discussion effectively interprets the results but can be more concise
We have carefully considered your suggestion to streamline the Discussion section. However, each point discussed is a result of extensive refinement through detailed responses to previous reviewer comments, which have emphasized the need for a thorough exploration of the complex issues presented. Our discussion is designed to comprehensively address these multifaceted aspects of the work, which we believe are crucial for supporting the study's findings and implications effectively.
We understand the importance of conciseness and have endeavored to make the section as direct as possible without omitting essential content. If specific areas appear redundant or overly detailed, we would appreciate further specific guidance on these sections.
- Conclusion:
- Highlight the broader implications of the findings.
- Mention the next steps for research or clinical application.
We had not included a specific Conclusions section after the Materials and Methods (as indicated by the journal template) as this was not mandatory and we already had a concluding paragraph at the end of the Discussion. This paragraph has now been expanded as suggested and moved to the ‘Conclusion’ section after the ‘Materials and Methods’, as follows:
Line 716-733:
- Conclusions
In conclusion, we present here a novel imaging method for the detection of pulmonary micrometastases in mice, by targeting activated pulmonary endothelium using a PET-detectable molecularly targeted contrast agent, [89Zr]-DFO-VCAM-MPIO. By retaining the microparticle platform used for our molecular MRI contrast agents, we both retain their clearance characteristics yielding highly specific target signal changes, and provide a dual modality agent that can be used to detect micrometastases in different tissue beds with the optimal imaging approach for each. The agent is in itself a platform, as the basic MPIO component can be conjugated to a range of different targeting and contrast molecules to create varied and multifunctional agents. Looking ahead, the next steps for research should focus on further refining the probe’s specificity and efficiency leading to clinical trials. Downstream, exploring the integration of this imaging technology into routine clinical practice will be crucial. On a research level, expanding the application of this probe to other types of cancers could provide comprehensive insights into its versatility and effectiveness across different oncological contexts. These steps will be pivotal in moving from experimental application to a standard component of oncological imaging and treatment planning, paving the way for advances in personalized medicine and improved prognosis for cancer patients.
- References
- References older than 20 years ago are only ideal incases of history, evolution etc.
The following references have been removed.
- Seo, J.B.; Im, J.G.; Goo, J.M.; Chung, M.J.; Kim, M.Y. Atypical Pulmonary Metastases: Spectrum of Radiologic Findings. Radi-ographics 2001, 21, 403–417, doi:10.1148/radiographics.21.2.g01mr17403.
- Hatabu, H.; Alsop, D.C.; Listerud, J.; Bonnet, M.; Gefter, W.B. T2* and Proton Density Measurement of Normal Human Lung Parenchyma Using Submillisecond Echo Time Gradient Echo Magnetic Resonance Imaging. European Journal of Radiology 1999, 29, 245–252.
- Kostakoglu, L.; Agress, H.; Goldsmith, S.J. Clinical Role of FDG PET in Evaluation of Cancer Patients. RadioGraphics 2003, 23, 315–340, doi:10.1148/rg.232025705.
- Anzai, Y.; Piccoli, C.W.; Outwater, E.K.; Stanford, W.; Bluemke, D.A.; Nurenberg, P.; Saini, S.; Maravilla, K.R.; Feldman, D.E.; Schmiedl, U.P.; et al. Evaluation of Neck and Body Metastases to Nodes with Ferumoxtran 10-Enhanced MR Imaging: Phase III Safety and Efficacy Study. Radiology 2003, 228, 777–788, doi:10.1148/radiol.2283020872.
Consequently, the following text revision and references have been added in the ‘Introduction’ section:
Line 43-47: Owing to the capacity of cancer cells to spread to the lungs, this is the second most frequent site of metastasis, with an estimated 20 to 54% of tumors originating elsewhere in the body metastasizing to this organ [3]. The most prevalent cancers that metastasize to the lung parenchyma include those originating from the breast, lung, colon, uterine leiomyosarcoma, and head and neck squamous cell carcinomas [3,4].
- Zhao, X.; Wen, X.; Wei, W.; Chen, Y.; Zhu, J.; Wang, C. Clinical Characteristics and Prognoses of Patients Treated Surgically for Metastatic Lung Tumors. Oncotarget 2017, 8, 46491–46497, doi:10.18632/oncotarget.14822.
- Stella, G.M.; Kolling, S.; Benvenuti, S.; Bortolotto, C. Lung-Seeking Metastases. Cancers (Basel) 2019, 11, 1010, doi:10.3390/cancers11071010.
Also, minor grammar changes have been made to the manuscript and are all highlighted.
